# Localized wastewater surveillance showed correlation but no early warning during Bengaluru's Omicron wave

Siva Athreya[1], Farah Ishtiaq[2], Tarun Khandelwal[3], Chitra Pattabiraman[3], Lakshminarayana Rao[4], Rajesh Sundaresan[5]*, Reshma Mohan Thattaramppilly[4]

1 International Centre for Theoretical Sciences - TIFR, Bengaluru, India, 2 Tata Institute for Genetics and Society, Bengaluru, India, 3 ARTPARK, Indian Institute of Science Campus, Bengaluru, India, 4 Centre for Sustainable Technologies, Indian Institute of Science, Bengaluru, India, 5 Electrical Communication Engineering, Indian Institute of Science, Bengaluru, India

* rajeshs@iisc.ac.in

## Abstract

Wastewater surveillance is an effective tool for monitoring the spread of infectious diseases such as COVID-19. In August 2021, a citywide surveillance effort was initiated in Bengaluru to analyze viral loads from 28 sewage treatment plants (STPs). The study found a strong correlation between aggregated viral loads and citywide COVID-19 case counts. However, the lack of localized clinical data limited the ability to assess infection trends at the STP level. In this follow-up study, we incorporate granular clinical data from 198 administrative units in the city. We find similar trends between viral loads at individual STPs and the cases in their catchment areas. A typical confidence interval for the Pearson correlation between clinical cases and wastewater viral loads at an STP is approximately $0.56$–$1.00$, based on the median bounds across the STPs; at the city-level it is $0.67$–$1.00$. However, our analysis shows no reliable indication of a lead or early warning—the viral loads and reported cases rise simultaneously. It is important to note that our study is limited to the first Omicron wave of the pandemic. To quantify lead time, we used correlation and change-point analysis. These results underscore the potential of localized wastewater surveillance for real-time monitoring but highlight its limitation in early outbreak detection.

## Author summary

We aimed to assess the effectiveness of wastewater surveillance in detecting COVID-19 trends at localized scales within Bengaluru during the first Omicron wave. To enable this localized study, we undertook two key tasks: first, we mapped the catchment areas supplying wastewater to the sewage treatment plants (STPs), and second, we incorporated clinical data from 198 administrative units, for the time period when SARS-CoV-2 viral loads were monitored. This mapping of catchment areas and integration of granular clinical data was not done in previous studies. We find that viral

**Data availability statement:** Data is available at http://dataportal.icts.res.in/wastewater/.

**Funding:** The research of SA is partially supported by the Knowledge Exchange Grant from ICTS-TIFR under project no. RTI4001 of the Department of Atomic Energy, Government of India. The research of RS and TK is partially supported by an IOE grant, CISCO-IISc Centre for Networked Intelligence (https://cni.iisc.ac.in/), and the Isaac Centre for Public Health at the Indian Institute of Science. Wastewater surveillance of SARS-CoV-2 was supported by funding from the Rockefeller Foundation (grant 2021 HTH 018) and the Indian Council of Medical Research grant to (FI) the Tata Institute for Genetics and Society and Tata Trusts. The funders had no role in study design, data collection and analysis, decision to publish, or preparation of the manuscript.

**Competing interests:** The authors have declared that no competing interests exist.

loads closely mirrored infection trends at a localized level; however, they did not provide an advance indication of cases, as they rose concurrently with reported infections.

## Introduction

Environmental surveillance of wastewater has proven to be highly effective in detecting diseases in populations, with previous applications including tracking poliovirus and emerging applications such as detecting antimicrobial resistance [1]. Wastewater surveillance entails continuous, systematic sampling, and computing viral concentrations. Interpretation of these data has helped public health officials understand the prevalence of the disease and design appropriate responses [2]. During the COVID-19 pandemic, wastewater surveillance has gained significant attention worldwide, particularly due to its potential to serve as an early warning system [1].

Multiple countries have set up wastewater surveillance systems for COVID-19. Data from countries such as the Netherlands, Spain, Australia, and the USA suggest that SARS-CoV-2 RNA can be detected in sewage in advance of clinical cases [3–10]. While initial studies and reports were limited to developed countries, there has been a growing body of work from urban centers in developing countries supporting the use of routine wastewater surveillance to monitor the COVID-19 pandemic including the Indian cities of Mumbai, Pune, Hyderabad, Ahmedabad, and Bengaluru [10–14]. There has also been work on mapping wastewater catchment areas and setting up SARS-CoV-2 surveillance systems in Bangladesh, Nepal, and Malawi [15,16]. In low-income settings, a large fraction of the sewage is untreated, and the catchment areas for these informal sewage networks are more challenging to map [15]. Mapping catchment areas is crucial to localized wastewater surveillance, as it enables the systematic identification of specific geographic regions contributing pathogens to the wastewater system. Further, in such low-income settings, it is also useful to identify bellwether sewershed sites for permanent monitoring and to scale-up to subsites during outbreak periods to identify hotspots [13].

Multiple studies show that wastewater data can be a leading indicator for daily new cases [8,14,17–24]. These studies have used methods such as correlation, cross-correlation, generalized additive models, and time series analysis [8,19–25]. Most of these studies have found correlation between viral loads in wastewater and disease indicators such as daily cases or hospitalizations [3,8,11,17,19–26]. Correlation is sensitive to viral shedding characteristics, sampling design, the sewerage network, the size and population of the catchment area. The correlation improves with an increase in sampling frequency [26,27], due to reduction in estimation errors. The lead times, i.e., the number of days by which the wastewater viral load trajectories lead the clinical cases trajectories, reported in different studies vary substantially, with some studies reporting short lead times of two days and others as long as 1–2 weeks [8,17,19–25,28]. The lead times are also influenced by testing policy, access to testing, and delays in reporting cases apart from shedding dynamics of the circulating viral variant.

The SARS-CoV-2 virus is shed in respiratory secretions and feces [29–33]. The *viral load* is the number of RNA copies per unit ml of wastewater. While shedding

characteristics may vary across viral variants, a meta-analysis of the Omicron variant found that viral loads peaked 4–6 days after symptom onset and declined by day 10 [29,32]. This is consistent with a higher correlation for SARS-CoV-2 RNA levels in sewage and new cases rather than active cases or cumulative cases [26]. Infected individuals with no or mild clinical symptoms who are not tested also contribute to the viral load.

Our main objective in this study is to quantify the correlation between observed wastewater viral loads at STPs and reported cases in Bengaluru, a city in India with a population of 14 million. We chose the period around the first Omicron wave, i.e., from 15 November 2021 to 31 January 2022. During this period the viral load data in wastewater samples from 26 sewage treatment plants (STPs) were available from [12] and due to comprehensive testing policy, daily cases data were available at the BBMP (now Greater Bengaluru Authority) website. We first map the catchment areas from Bengaluru for each of these STPs. We then perform correlation analysis and implement a refined change-point detection analysis to evaluate if viral load leads cases in the city during the Omicron wave.

## Methodology

Bruhat Bengaluru Mahanagara Palike (BBMP) is Bengaluru's administrative body that manages its various civic amenities and services. Bengaluru comprises 198 BBMP *wards*, which are its local administrative units. The average ward population is approximately 60,000 individuals [34]. The city produces approximately 1,440 million liters per day (MLD) of wastewater, of which about 85% is treated in 26 STPs. The STPs vary in capacities (measured in MLD) and cover over 183 out of 198 wards (Fig 1). They are designed to treat and process wastewater generated from residential, commercial, and industrial sources. They are spread across the city and are operated by the Bengaluru Water Supply and Sewerage Board (BWSSB).

As stated earlier, our main objective is to understand the correlation between observed wastewater viral loads at STPs and reported cases. We chose the period around the Omicron wave for the study because data for both wastewater viral loads and daily cases were available due to diligent testing.

The wastewater surveillance in Bengaluru started in August, 2021. Samples were collected from STPs using the 'grab' method and tested for the presence of SARS-CoV-2. The data comprises viral load values in copies/ml from 26 STPs. For the period post January 4, 2022 data is also available as part of the supplementary data in [12]. Sampling was conducted every 10 days before January, 2022 and weekly thereafter. Henceforth, we refer this as 'weekly' for simplicity. S3 Table shows the number of samples for each STP during the study period.

BBMP collects and records the case data. The ward-level daily case numbers were reported at the BBMP website. Each case is attributed to the patient's ward of residence. The case data on the BBMP website was originally in PDF format. Upon request, the data was shared with us in machine-readable format. City-wide cases are calculated as the sum of cases reported across all wards.

### Mapping STP catchments

Out of 198 wards, 42 wards were mapped manually using the procedure described below. Mapping for the rest of the wards was given by BWSSB. Manually mapping wards to the respective STPs relied on utilizing a Geographic Information System (GIS) tool known as QGIS (version 3.32.2). The required input files included the geographical and administrative information of the wards in the Bengaluru Urban District, the location and capacity of each STP, and the sewer drainage network map. The information about the 198 wards in Bengaluru was compiled into a geopackage file (.gpkg), which was then imported as a layer into the QGIS project. The locations of the 26 STPs in Bengaluru were sourced from the Open City Urban Data Portal in KML file format and integrated into the QGIS project (Fig 1). The accuracy of the STP locations was cross-checked using a geo-referenced raster file obtained from BWSSB.

The sewer system in the Bengaluru Urban District has been designed in alignment with the natural terrain, utilizing the existing ridges and valleys. This sewerage network encompasses approximately 229 km$^2$ and comprises five primary drainage regions which are commonly referred to as Vrishbhavathi, Arakavathi, Hebbal, Koramangala & Challaghatta, and

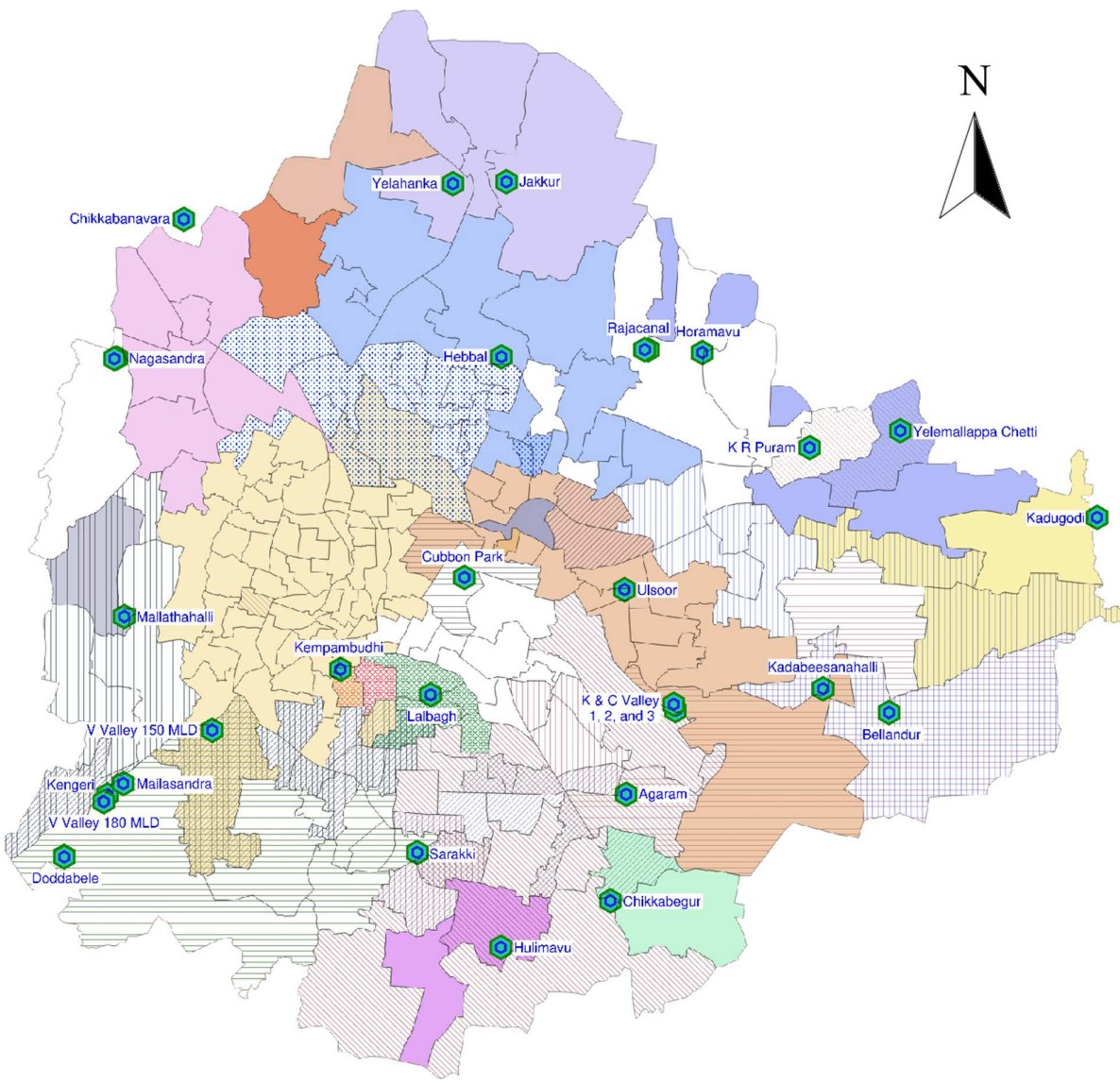

**Fig 1. The map of Bengaluru, STP locations, and ward boundaries.** Wards with the same texture send their wastewater to the same set of STPs. The samples were collected from the STPs shown. Wards that are not covered by testing are shown in white. Base layer: Bengaluru BBMP wards [35,36], Bengaluru City STPs [37,38]. License: Open Data Commons Open Database License (ODbL) v1.0.

Suvarnamukh (Fig 2). The sewer line maps in KML format with pipe diameters greater than 300 mm and less than 300 mm were obtained from the Open City Urban Data Portal. This data was also cross-checked with sewerage line maps (georeferenced raster files) obtained from the BWSSB website. A sewer line between two locations makes no distinction between the source and destination. We infer the source and destination from the knowledge of the terrain. The flow direction of sewage from each ward was determined based on the elevation of the valleys. Similarly, an assessment of drainage line density in each ward was conducted, facilitating the mapping of each ward to the corresponding STP. For instance, we infer that Sanjay Nagar ward sends its wastewater to Hebbal STP by inspecting the sewer lines (see S1 Fig. for the map). S2 Fig. shows the sewer line map of the entire city. To the best of our knowledge, this is the first focused

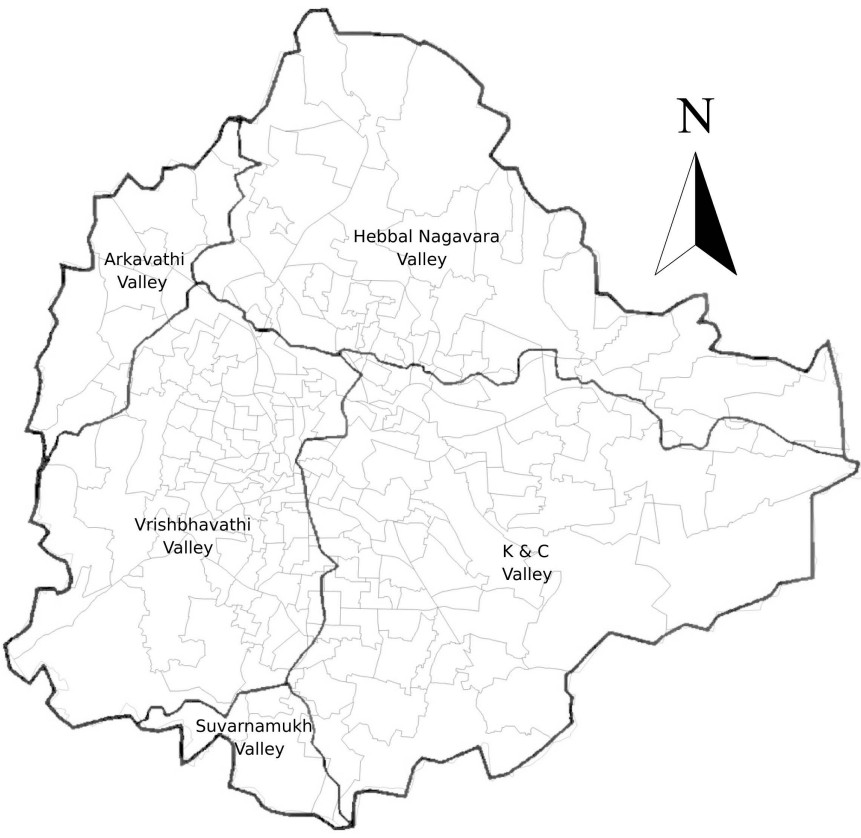

**Fig 2. The Valleys of Bengaluru.** The shapefile for the valleys of Bengaluru was created in QGIS (QGIS.org 3.32.2 'Lima' 2023) using a georeferenced raster file from literature [39]. Base layer: Bengaluru BBMP wards [35,36]. License: Open Data Commons Open Database License (ODbL) v1.0.

effort to map each ward to its STPs in the Bengaluru Urban District. The result of this mapping is given in Section Wards in the catchment of an STP.

**Pre-processing of viral loads**

Wastewater samples were collected at the inlets of the STPs to measure the viral load at each site. We interpolate the sparse data (based on actual dates of collection) to obtain *daily* viral loads. We use quadratic interpolation to estimate missing values in our data. We implement this using the interpolate() function from the Pandas Python module. If the interpolation outputs any negative values for viral loads, we project it to the nearest value on the set of valid viral loads, that is, we set them to zero.

For an STP $S$, we denote its viral load (possibly interpolated) on day $i$ as $v_S(i)$.

To compute citywide viral load, we combined the viral loads using the following formula:

$$v(i) = \frac{\sum\limits_{S \in \mathcal{S}} \left( v_S(i) \cdot \mathrm{MLD}_S \right)}{\sum\limits_{S \in \mathcal{S}} \mathrm{MLD}_S},$$

(1)

where $\mathcal{S}$ is the set of STPs and $MLD_S$ is the capacity of STP $S \in \mathcal{S}$. In other words, the viral load is calculated as the weighted average of the viral loads from all the STPs, with weights corresponding to their capacities.

In the above, we use STP capacity rather than daily or weekly flow-rate averages because (1) it represents the hydraulic and organic load the STP is engineered to treat while maintaining the effluent discharge standards, and (2) all STPs are operating continuously at full capacity and the city generates more sewage than the STP capacity.

We also combine viral loads from multiple STPs if they have a common catchment area, i.e., if the STPs receive wastewater from the same set of wards. This happens at two locations. Rajacanal (80 MLD) and Horamavu (20 MLD) are two STPs located close to each other that have a common catchment area. Hence, we treat the pair as a single STP and replace their daily viral loads with their weighted average. Similarly, Nagasandra (40 MLD) is a combination of two STPs: Nagasandra Old (20 MLD) and Nagasandra New (20 MLD). See S1 Text for an illustrative example.

## Pre-processing of case counts

The catchment area of an STP may not align perfectly with ward boundaries. To relate viral loads to case counts, we would ideally require cases to be reported at the catchment level. However, since cases are reported at the ward level, we estimate the number of cases within each STP's catchment area. We describe how we estimate below.

For a ward that sends wastewater to multiple STPs, we divide the cases from that ward equally (a simplifying assumption) among the STPs. For an STP $S$, the case count on a day $i$ is defined as:

$$c(i) = \sum_{j=1}^{K_S} \frac{\tilde{c}_j(i)}{k_j},$$

(2)

where $K_S$ is the number of wards in the catchment area of STP $S$, $k_j$ denotes the number of STPs into which the $j$-th ward drains, and $\tilde{c}_j(i)$ denotes the number of cases on day $i$ in the $j$-th ward. We calculate the case counts from 15 November 2021 to 31 January 2022.

## Computing correlations

Previous studies have reported linear and non-linear associations between viral loads in wastewater and COVID-19 case counts in catchment areas. For instance, linear, log-linear and non-linear correlations have been reported in [27,40] and [41], respectively. In our case, to determine the relationship between viral load and COVID-19 cases, we first plotted the data from all STPs on a scatter plot. The data revealed a linear trend, consistent across STPs. S3 Fig. shows an example using four STPs. Viral load is expected to be proportional to cases because, in a constant population, an increase in infected individuals correlates with more viral copies being shed into the wastewater. If the number of cases doubles, the total viral load in wastewater should also double, assuming similar viral shedding rates [29,31]. Pearson correlation is then used to quantify the strength of the linear relationship, providing a robust measure of how closely variations in viral load align with changes in case numbers.

Let $\mathbf{x} = (x(1), \ldots, x(n))$ and $\mathbf{y} = (y(1), \ldots, y(n))$ be vectors of length $n$. The Pearson correlation between $\mathbf{x}$ and $\mathbf{y}$, denoted as $\rho_{\mathbf{xy}}$, is given by the formula:

$$\rho_{\mathbf{xy}} = \frac{\sum_{i=1}^{n} (x(i) - \bar{x})(y(i) - \bar{y})}{\sqrt{\sum_{i=1}^{n} (x(i) - \bar{x})^2 \sum_{i=1}^{n} (y(i) - \bar{y})^2}}$$

(3)

where $\bar{x}$ and $\bar{y}$ are the mean values of the vectors $\mathbf{x}$ and $\mathbf{y}$, respectively.

A Pearson correlation value of 1 indicates a perfect positive correlation, -1 indicates a perfect negative correlation, and 0 indicates no correlation.

We now see how to compute the correlation between cases and viral loads. We begin by defining the following terms for a given Sewage Treatment Plant (STP), $S$:

- **Time period $T$**: Let $T$ denote the dates from 15 November 2021 to 31 January 2022. There are $n = 77$ days in this period.

- **Shift parameter $\eta$:** The parameter $\eta$ is an integer between -14 and 14 that indicates a shift in the time period $T$. $T_\eta$ refers to the 77-day period starting from $\eta$ days before 15 November 2021. For example, $T_3$ denotes the dates from 12 November 2021 to 28 January 2022. Similarly, $T_{-2}$ denotes the dates from 17 November 2021 to 2 February 2022.

- **Case Vector ($\mathbf{c} = (c(1), \ldots, c(n))$)**: The vector representing the daily COVID-19 cases reported within the catchment area of $S$ during the period $T$. The number of cases on day $i$ is calculated according to Eq. 2.

- **Interpolated Daily Viral Load Vector ($\tilde{\mathbf{v}}_\eta$)**: The vector obtained by interpolating viral load measurements taken weekly from $S$ using quadratic interpolation. The viral loads are for the days in the period $T_\eta$.

- **True Viral Load Vector ($\mathbf{v}_\eta$)**: The hypothetical vector of the true daily viral load values we would have obtained if measurements had been made daily during $T_\eta$. This is the vector that $\tilde{\mathbf{v}}_\eta$ approximates.

When calculating the correlation between cases and viral loads, we keep the case vector fixed, i.e., the case vector is always for the period $T$. The viral load vector is shifted relative to the case vector for $\eta \in \{-14, \ldots, 14\}$. We are interested in the maximum correlation over all shifts. The maximum correlation $\rho_{\mathbf{c}\tilde{\mathbf{v}}}$ and the corresponding shift $\eta^\star$ are defined as:

$$\rho_{\mathbf{c}\tilde{\mathbf{v}}} := \max_{\eta \in \{-14,\ldots,14\}} \rho_{\mathbf{c}\tilde{\mathbf{v}}_\eta} \quad \text{and} \quad \eta^\star := \operatorname*{argmax}_{\eta \in \{-14,\ldots,14\}} \rho_{\mathbf{c}\tilde{\mathbf{v}}_\eta}. \tag{4}$$

Table 2 (columns 2 and 3) shows the values of the above quantities for each STP. A positive value of $\eta^\star$ indicates that daily viral loads lead the cases.

### Validating the sample size and interpolation method

The wastewater samples were collected weekly during the study period. Consequently our correlation analysis is on interpolated viral data based on 8–9 samples collected per STP. Thus, there is a need to validate the interpolation method. Ideally, one would estimate the information loss by comparing the interpolated values with the daily viral loads (ground truth). Since daily viral loads are not available for STPs in Bengaluru, we use two indirect ways to justify the interpolation method.

There were six STPs in California where daily viral load measurements were made during the Omicron period ([42,43]). As mentioned earlier, the BBMP also reported daily cases ward wise during the Omicron period. We validate the interpolation method using both the datasets described above. We explain in detail the procedure for the STPs in California and the same will apply for the daily cases data from BBMP. For each of the STPs, we apply the following procedure:

**Step 1: Ground truth** We select the viral load data from 11 weeks during the Omicron wave. We denote this sequence by $\nu$.

**Step 2: Weekly samples** From $\nu$, we retain only those viral loads where the collection day is a Monday. The rest of the values are marked as 'missing'. We denote this sequence by $\nu_\mathbf{w}$.

**Step 3: Interpolation** We apply quadratic interpolation to $\nu_\mathbf{w}$ to fill up the 'missing' values. Thus, we get daily interpolated values for 77 days (11 weeks). We refer to this sequence as $\tilde{\nu}$.

**Step 4: Correlation** We compare the results of the interpolation method with the ground truth. In particular, we find the correlation coefficient between $\nu$ and $\tilde{\nu}$.

The results of this exercise are shown in S2 Table. The third column shows the correlation between viral load and the interpolated values. The predominantly high values of correlation indicate the effectiveness of interpolation. We also

calculate the uncertainty in the viral load estimates. The difference between the estimated viral load and the actual viral load during the baseline period will be much smaller than during the wave. Hence, instead of computing the absolute errors, we use error factors. The error factor is the ratio of the estimated (interpolated) value and the true viral load. During the baseline period, viral load values that are close to zero can lead to high error factors, which skew the distribution. Hence, we first remove the outliers before computing standard error. S4 Fig. shows the weekly mean errors. The y-axis shows the error factor. Each point represents the mean error factor for a given week, and the vertical whiskers show the standard deviation, indicating the variability of the error factor across all points within that week.

We also used the above procedure outlined in Steps 1–4 (above) to accurately measure information loss due to interpolation on the BBMP cases data. For each STP in Bengaluru, we consider the case vector **c** as the ground truth. We retain the cases only on those days when wastewater was sampled in the corresponding STP. We denote this sequence of 8–10 data points as $c_w$. We apply quadratic interpolation to $c_w$ to get $\tilde{c}$ and then find the correlation coefficient between **c** and $\tilde{c}$.

The results of this exercise are shown in S3 Table. Even when we pick only 8–10 data points, we see a high correlation (>90%) between the interpolated values and the daily case counts. S5 Fig. shows the error bars for case estimates.

The weekly error bars in S4 Fig. and S5 Fig. are not directly used in the calculation. However, based on the error bars shown in the figures, we assume that the interpolation method introduces about 25% standard deviation error in each viral load value. This assumption is used in Step 2 below.

We next quantify the uncertainties introduced by the interpolation process while calculating the correlations between cases and viral loads. We find the confidence interval of true correlation $\rho_{cv}$ for each STP in three steps:

**Step 1: Calculate $\rho_{c\tilde{v}}$.** We calculate the correlation between the case vector **c** and the interpolated viral load $\tilde{v}$. This value is shown in the second column of Table 2.

**Step 2: Estimate $\rho_{v\tilde{v}}$.** We see that interpolation perturbs the values of the original vector by about 25% (S4 Fig. and S5 Fig.). Hence, the question of estimating $\rho_{v\tilde{v}}$ can be re-formulated as estimating the correlation between vectors **v** and $\mathbf{v} + 0.25\mathbf{v} \odot \mathbf{z}$ where **z** is a vector of independent and identically distributed standard normal values and $\odot$ is component-wise multiplication. We find, via simulation, that the correlation is concentrated around 0.895 with a standard deviation of 0.029 (S6 Fig.). Hence, we use

$$[0.895 - 2 \times 0.029, 0.895 + 2 \times 0.029] = [0.837, 0.953]$$

as the 95% confidence interval for $\rho_{v\tilde{v}}$. Note that this interval is same for all STPs. Our estimate of $\rho_{v\tilde{v}}$ is reasonable given the correlation values for viral loads found in California STPs (S2 Table).

**Step 3: Estimate $\rho_{cv}$.** We bound the value of the true correlation $\rho_{cv}$ in terms of the quantities $\rho_{v\tilde{v}}$ and $\rho_{c\tilde{v}}$. From the previous step, we have $\rho_{v\tilde{v}} \in [0.837, 0.953]$ with 95% confidence. To bound the value of true correlation, we use the following inequality that relates pairwise correlations between three vectors proved in [44]:

$$\rho_{v\tilde{v}}\rho_{c\tilde{v}} - \sqrt{1 - \rho_{v\tilde{v}}^2}\sqrt{1 - \rho_{c\tilde{v}}^2} \leq \rho_{cv} \leq \rho_{v\tilde{v}}\rho_{c\tilde{v}} + \sqrt{1 - \rho_{v\tilde{v}}^2}\sqrt{1 - \rho_{c\tilde{v}}^2}$$

The above inequality provides a range of values for $\rho_{cv}$ for fixed values of $\rho_{v\tilde{v}}$ and $\rho_{c\tilde{v}}$. By using the 95% confidence interval for $\rho_{v\tilde{v}}$, i.e., [0.837,0.953], we get a similar confidence interval for $\rho_{cv}$ which is shown in the last column of Table 2.

### Change-point detection using Cumulative SUM (CUSUM)

Does a rise in viral load at an STP predict an impending increase in cases? We used a change-point detection approach to answer this question. A change-point detector monitors sequential data for changes in the statistical properties of the data. A good detector quickly detects a change while reducing the chance of a false alarm. Our interest is in both kinds of sequential data: viral loads at an STP and confirmed cases in its catchment area.

We describe the general change-point detection problem. Suppose the data points $y_1, \ldots, y_n$ are obtained sequentially. Suppose we are also given a pre-change (reference) distribution $f_0$ and a post-change distribution $f_1$. We wish to determine $\tau$, the change-point location, where $y_s$ at time $s$ is distributed (notation '$\sim$') according to $f_0$ or $f_1$ as follows:

$$\begin{aligned} y_s &\sim f_0 \quad \text{if } 0 \leq s \leq \tau \\ y_s &\sim f_1 \quad \text{if } \tau < s \leq n. \end{aligned}$$

The distributions $f_0$ and $f_1$ will reflect the statistical properties of the viral loads (or cases) during the period before the wave and at the start of the wave, respectively. At the start of the wave, we do expect a trended change, but to fix ideas, assume a constant post-change distribution. There are two standard performance metrics to evaluate based on the two types of errors that can occur. Firstly, the detector can raise a false alarm when there is no change in the sequential data. This is quantified by average run length (ARL), the expected time until a change is detected when the given sequence comes only from $f_0$. Secondly, there could be a delay in detecting a change. This is quantified by worst-case average detection delay (ADD) as defined by Lorden [45].

We use the cumulative sum algorithm (CUSUM) introduced by Page [46]. This is a (generalized) likelihood-ratio test that assumes that distributions $f_0$ and $f_1$ are known, but the change-point is unknown. Given the sequential data $y_1, \ldots, y_n$, CUSUM computes the cumulative log-likelihood ratio considering an unknown change-point location $\tau$. The test statistic $Z_n$ is defined as:

$$Z_n := \max_{1 \leq \tau \leq n} \sum_{s=\tau}^{n} \log \frac{f_1(y_s)}{f_0(y_s)} \quad \text{for } n \geq 0$$

The test statistic is easy to compute via a recursive formula:

$$Z_n = \left( Z_{n-1} + \log \frac{f_1(y_n)}{f_0(y_n)} \right)^+, \quad Z_0 = 0$$

The CUSUM procedure raises an alarm the first time $Z_n$ exceeds a pre-defined threshold $h$. The ARL of CUSUM depends on the threshold $h$ (ARL $\propto \exp(h)$), and once the ARL is calibrated, the CUSUM algorithm minimizes the ADD in the asymptotic sense [47]. For this reason, CUSUM is an asymptotically optimal algorithm for hypothesis testing when the distributions $f_0$ and $f_1$ are known.

The CUSUM algorithm cannot be directly applied to the viral load or case data. The reason is the dependence on time for both cases and viral loads. During post-change, the cases and viral loads increase with time. We choose a different test statistic to remove this dependence on time and embed the current setting within the framework of CUSUM. We set

$$y_i = v(i + 1) - v(i),$$

for each $i$. Note that $y_i$ denotes the rate of change of the viral load on the $i$th day. Although $y_i$'s are 'detrended' they are not independent. In particular, for each $i$, $y_{i+1}$ and $y_i$ are correlated since both depend on $v(i)$. To ensure independence, we use $y_k$ for all odd $k$ as they are mutually independent.

The pre-change distribution $f_0$ and post-change distribution $f_1$ of the $y_k$s are assumed to be normally distributed with mean $\mu_0$ and $\mu_1$ and standard deviation $\sigma_0$ and $\sigma_1$, respectively. We estimate the parameters of the pre-distribution $f_0$ from the values (cases or viral loads) between 15 Nov 2021 and 24 Dec 2021. We estimate the parameters of the post-distribution $f_1$ from the values between 25 Dec 2021 and 14 Jan 2022 (S1 Table). This period corresponds to the rise of

the Omicron wave. The table shows that both the viral and case distributions have means close to zero during the period pre-change and switch to positive values in the post-change phase.

Assuming the parameter values in S1 Table, we run the CUSUM algorithm to detect change points in both viral load and case data over the same period. We are interested in knowing if the CUSUM detects the change in viral loads earlier than it does with the cases. We say the viral load *leads* the cases if this happens. To quantify the lead time we first define a few terms. The CUSUM algorithm raises an alarm when it detects a change point, either in cases or viral loads and terminates after that.

**Alarm dates.** Let Cases-CP (cases change point) denote the date the CUSUM algorithm raises an alarm based on case data. Let VCPD be the alarm date based on daily viral load. The lead time Lead-VCPD is the temporal difference between the alarm dates of the two sequences:

$$\text{Lead-VCPD} = \text{Cases-CP} - \text{VCPD}$$

A positive lead time (Lead-VCPD >0) indicates that the alarm date VCPD precedes the alarm date Cases-CP. For example, if VCPD = 2021-12-24 and Cases-CP = 2021-12-30, then:

$$\text{Lead-VCPD} = 2021\text{-}12\text{-}30 - 2021\text{-}12\text{-}24 = 6 \text{ days,}$$

i.e., the daily viral load data provides a 6-day early warning signal. We also define an alternate notion of lead time. The earliest sample collection day after VCPD is called VCPC (change-point based on collection date). Consider an example given in Fig 3. Suppose the samples were collected on 2nd, 9th and 16th day of the month. Hence, viral loads are available only for three days. Suppose the CUSUM algorithm detects a change-point on the 12th day of the month. In reality, no viral load is measured on this day. In practice, the actual rise would not be detected until the next collection day, i.e., the 16th of the month. We define the lead time based on VCPC as:

$$\text{Lead-VCPC} = \text{Cases-CP} - \text{VCPC}$$

The viral load is said to provide an *early warning* if Lead-VCPC >0.

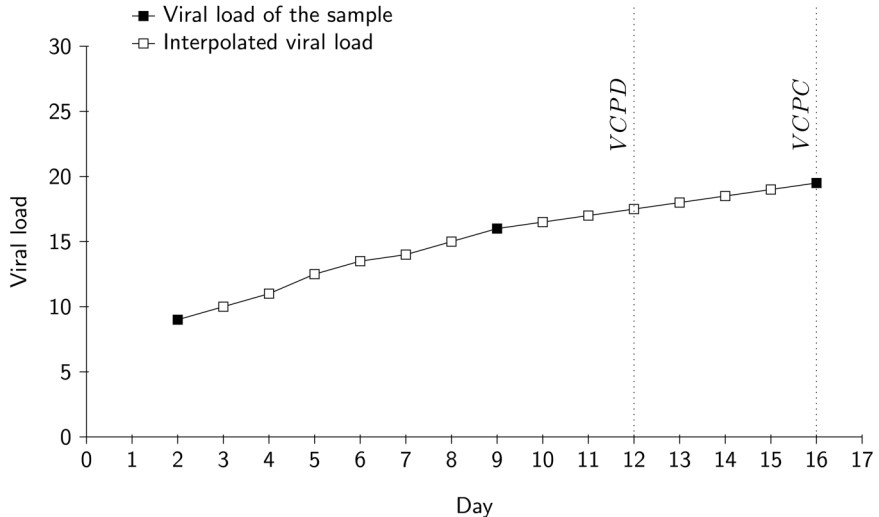

**Fig 3. Illustration of the difference between VCPD and VCPC.** The change-point based on interpolated daily viral load is detected on day 12. Hence, VCPD is the 12th of the month. The earliest sample collection day after detection is day 16. Hence, VCPC is 16th of the month.

## Results

### Wards in the catchment of an STP

We identified the wards within the catchment area of each STP, following the methodology given in Section Mapping STP catchments). The details are in Table 1 and visualized in Fig 1. The mapping allows us to attribute the cases in the catchment area to the corresponding STP. Also, this mapping may be of independent interest for the localized surveillance of other pathogens, such as Influenza and Mpox.

### Correlation analysis

The results of the correlation analysis are summarized in Table 2. For each STP, we have the maximum correlation value of cases and daily viral loads and the corresponding shift $\eta^\star$. The median shift is three days. At first glance, it suggests that viral load leads the cases by three days. The observed three-day lead in the interpolated daily viral load data is an artifact of the interpolation process, rather than a true reflection of the relationship between viral loads and case numbers. Since viral load measurements are taken weekly, interpolating this data to a daily frequency introduces a lead that may not accurately represent the actual timing between viral loads and cases. Therefore, this lead should not be considered significant. A lead of seven or more days, however, would be more meaningful, as it aligns with the weekly sampling schedule and could provide useful information for early warning.

### Change-point analysis

Correlation being a retrospective analysis, we adapted a classical change-point analysis on the viral load and cases time series to see if there was an early warning. Table 3 shows the alarm dates for cases and viral loads across all STPs. Leads are measured based on VCPD and VCPC dates (with threshold $h = 20$). We consider the lead time based on collection dates more accurate, so the entries are sorted according to VCPC lead times. As an analogous indication of statistical power, we vary the threshold $h$ from 10 to 20 and tabulate the mean and standard deviations for lead times (Table 4).

## Discussion

We use the well-known standard correlation and change-point analysis at the STP level (as opposed to citywide aggregate) due to the mapping of wards to the respective STPs using a Geographic Information System (GIS) tool kit, daily cases data from BBMP, and detailed drainage maps from BWSSB (Fig. 1). From the data we estimated the catchment area associated to each STP, providing a basis for correlating the ward cases with the respective STP viral load. Apart from 3-4 STPs, during the wave, there is high (>0.80) correlation between the observed viral load and the cases observed in the respective catchment area (Table 2). The correlation in Yelahanka is low (0.48) due to a low viral load value between the initial rising phase and the peak; the correlation in Lalbagh and Bellandur is moderate ($0.79 - 0.80$) again due to low viral loads but post peak; we do not know the reasons for these low values. The correlation in Jakkur being moderate (0.79) can be attributed to missing data values post the wave's peak.

In our 77 day window, we had 8–9 data points per STP. We then used interpolation to derive daily samples from weekly wastewater data to determine the correlation with the daily cases. We notice a high correlation between the interpolated and actual values of viral load during during the Omicron wave (when there is a well defined peak along with a clear rise and fall in viral loads). However we observe a dip in correlation between actual and interpolated values during baseline period with stable but low viral loads. This is also observed on viral load data from STPs in California, where over the three month period from Nov 2022 to Dec 2023 the method yielded correlations of around 20–30% but with much higher correlations during the Omicron wave (S2 Table). We have also addressed the impact of having 8–9 data points and have quantified the interpolation error and uncertainty in correlation obtained in the Section Validating the sample size and

**Table 1. STPs and the list of wards in their catchment.**

| STP-Ward mapping |
| --- |
| Agaram (35 MLD) |
| 114-Agaram, 125-Marenahalli, 152-Suddagunte Palya, 168-Pattabhiram Nagar, 169-Byrasandra, 170-Jayanagar East, 171-Gurappanapalya, 172-Madivala, 174-HSR Layout, 175-Bommanahalli, 179-Shakambari Nagar, 187-Puttenahalli, 188-Bilekhalli, 189-Hongasandra, 192-Begur, 193-Arakere, 196-Anjanapura |
| Bellandur (90 MLD) |
| 85-Dodda Nekkundi, 86-Marathahalli, 149-Varthuru, 150-Bellanduru, 174-HSR Layout |
| Chikkabanavara (5 MLD) |
| 11-Kuvempu Nagar |
| Chikkabegur (5 MLD) |
| 190-Mangammanapalya, 191-Singasandra |
| Cubbon Park (4 MLD) |
| 93-Vasanth Nagar, 110-Sampangiram Nagar |
| Doddabele (60 MLD) |
| 71-Hegganahalli, 72-Herohalli, 129-Jnana Bharathi ward, 130-Ullalu, 154-Basavanagudi, 159-Kengeri, 160-Rajarajeshwari Nagar, 161-Hosakerehalli, 162-Girinagar, 164-Vidyapeeta ward, 165-Ganesh Mandir ward, 166-Karisandra, 167-Yediyur, 182-Padmanabha Nagar, 183-Chikkalsandra |
| Halasuru (2 MLD) |
| 59-Maruthi Seva Nagar, 79-Sarvagna Nagar |
| Hebbal (100 MLD) |
| 16-Jalahalli, 17-J P Park, 18-Radhakrishna Temple Ward, 19-Sanjaya Nagar, 20-Ganga Nagar, 21-Hebbala, 22-Vishwanath Nagenahalli, 31-Kushal Nagar, 33-Manorayanapalya, 34-Gangenahalli, 35-Aramane Nagara, 36-Mattikere, 37-Yeshwanthpura, 42-Lakshmi Devi Nagar, 45-Malleswaram, 46-Jayachamarajendra Nagar, 48-Muneshwara Nagar |
| Henuur (1 MLD) |
| 23-Nagavara |
| Hulimavu (10 MLD) |
| 193-Arakere, 194-Gottigere |
| Jakkur (15 MLD) |
| 1-Kempegowda Ward, 2-Chowdeswari Ward, 4-Yelahanka Satellite Town, 5-Jakkuru |
| K & C Valley (150 MLD) |
| 109-Chickpete, 110-Sampangiram Nagar, 115-Vannarpet, 116-Nilasandra, 117-Shanthi Nagar, 118-Sudham Nagara, 120-Cottonpete, 139-K R Market, 143-Vishveshwara Puram, 144-Siddapura, 145-Hombegowda Nagara, 146-Lakkasandra, 147-Adugodi, 148-Ejipura, 151-Koramangala |
| K & C Valley (218 MLD) |
| 58-New Tippasandara, 59-Maruthi Seva Nagar, 60-Sagayarapuram, 61-S K Garden, 62-Ramaswamy Palya, 63-Jayamahal, 78-Pulikeshinagar, 79-Sarvagna Nagar, 80-Hoysala Nagar, 87-HAL Airport, 88-Jeevanbhima Nagar, 89-Jogupalya, 90-Halsoor, 91-Bharathi Nagar, 92-Shivaji Nagar, 93-Vasanth Nagar, 112-Domlur, 113-Konena Agrahara, 150-Bellanduru |
| K & C Valley (30 MLD) |
| 148-Ejipura |
| K & C Valley (60 MLD) |
| 170-Jayanagar East, 171-Gurappanapalya, 172-Madivala, 173-Jakkasandra, 174-HSR Layout, 175-Bommanahalli, 176-BTM Layout, 177-J P Nagar, 178-Sarakki, 187-Puttenahalli, 190-Mangammanapalya, 195-Konankunte |
| K R Puram New (20 MLD) |
| 25-Horamavu, 26-Ramamurthy Nagar, 51-Vijnanapura |
| K R Puram Old (20 MLD) |
| 52-K R Puram, 53-Basavanapura |
| Kadabeesanahalli (50 MLD) |
| 50-Benniganahalli, 56-A Narayanapura, 57-C V Raman Nagar, 81-Vijnana Nagar, 82-Garudachar Playa, 84-Hagadur, 86-Marathahalli, 149-Varthuru |
| Kadugodi (6 MLD) |
| 82-Garudachar Playa, 83-Kadugodi, 84-Hagadur |
| Kempambudhi (1 MLD) |
| 142-Sunkenahalli, 155-Hanumanth Nagar |

*(Continued)*

**Table 1.** (Continued)

| STP-Ward mapping |
| --- |
| Kengeri (60 MLD) |
| 154-Basavanagudi, 159-Kengeri, 160-Rajarajeshwari Nagar, 161-Hosakerehalli, 162-Girinagar, 164-Vidyapeeta ward, 165-Ganesh Mandir ward, 166-Karisandra, 167-Yediyur, 182-Padmanabha Nagar, 183-Chikkalsandra |
| Lalbagh (1.5 MLD) |
| 143-Vishveshwara Puram, 144-Siddapura, 153-Jayanagar, 167-Yediyur |
| Madivala (4 MLD) |
| 176-BTM Layout, 177-J P Nagar |
| Mailasandra (75 MLD) |
| 160-Rajarajeshwari Nagar, 180-Banashankari Temple ward, 181-Kumaraswamy Layout, 184-Uttarahalli, 185-Yelchenahalli, 186-Jaraganahalli, 197-Vasanthpura, 198-Hemmigepura |
| Mallathahalli (5 MLD) |
| 72-Herohalli |
| Nagasandra (40 MLD) |
| 12-Shettihalli, 13-Mallasandra, 14-Bagalakunte, 15-T Dasarahalli, 38-HMT Ward, 39-Chokkasandra, 41-Peenya Industrial Area, 70-Rajagopal Nagar |
| Rajacanal (80 MLD) and Horamavu (20 MLD) |
| 6-Thanisandra, 7-Byatarayanapura, 8-Kodigehalli, 9-Vidyaranyapura, 10-Dodda Bommasandra, 24-HBR Layout, 27-Banasavadi, 28-Kammanahalli, 29-Kacharkanahalli, 30-Kadugondanahalli, 31-Kushal Nagar, 32-Kaval Bairasandra, 47-Devara Jeevanahalli, 48-Muneshwara Nagar, 78-Pulikeshinagar |
| Sarakki (5 MLD) |
| 147-Adugodi, 148-Ejipura, 151-Koramangala, 173-Jakkasandra, 178-Sarakki, 187-Puttenahalli, 195-Konankunte |
| V Valley (330 MLD) |
| 35-Aramane Nagara, 43-Nandini Layout, 44-Marappana Palya, 45-Malleswaram, 64-Rajamahal Guttahalli, 65-Kadu Malleshwar Ward, 66-Subramanya Nagar, 67-Nagapura, 68-Mahalakshimpuram, 69-Laggere, 73-Kottegepalya, 74-Shakthi Ganapathi Nagar, 75-Shankar Matt, 76-Gayithri Nagar, 77-Dattatreya Temple, 92-Shivaji Nagar, 94-Gandhinagar, 95-Subhash Nagar, 96-Okalipuram, 97-Dayananda Nagar, 98-Prakash Nagar, 99-Rajaji Nagar, 100-Basaveshwara Nagar, 101-Kamakshipalya, 102-Vrisabhavathi Nagar, 103-Kaveripura, 104-Govindaraja Nagar, 105-Agrahara Dasarahalli, 106-Dr. Raj Kumar Ward, 107-Shivanagara, 108-Sriramamandir, 109-Chickpete, 120-Cottonpete, 121-Binnipete, 122-Kempapura Agrahara, 123-Vijayanagar, 124-Hosahalli, 125-Marenahalli, 126-Maruthi Mandir ward, 127-Mudalapalya, 128-Nagarabhavi, 131-Nayandahalli, 132-Attiguppe, 133-Hampi Nagar, 134-Bapuji Nagar, 135-Padarayanapura, 136-Jagajivanaramnagar, 137-Rayapuram, 138-Chalavadipalya, 140-Chamrajpet, 141-Azad Nagar, 154-Basavanagudi, 155-Hanumanth Nagar, 156-Srinagar, 157-Gali Anjenaya Temple ward, 158-Deepanjali Nagar, 160-Rajarajeshwari Nagar, 163-Katriguppe |
| Yelahanka (10 MLD) |
| 3-Atturu |
| Yellamallappachetty (15 MLD) |
| 53-Basavanapura, 54-Hudi, 55-Devasandra |

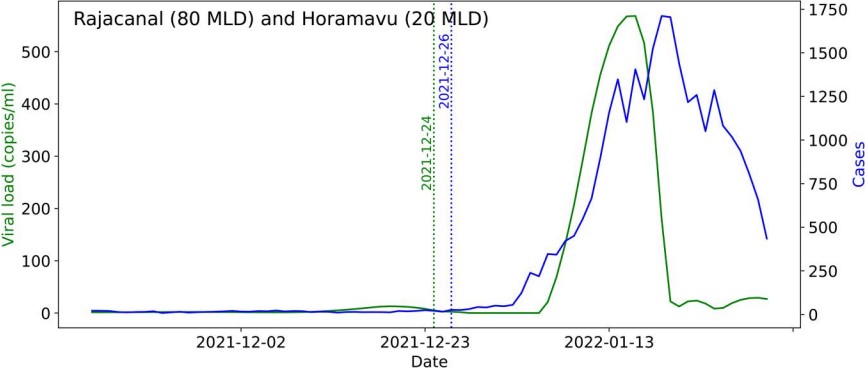

**Fig 4. The output of CUSUM on (composite of) Rajacanal and Horamavu STPs.** Viral loads and cases increased over time during the post-change period. The blue dotted line shows the change-point for cases and the green dotted line shows the change-point for viral loads (VCPC). CUSUM raised an alarm for viral loads two days before it raised an alarm for cases.

**Table 2. Correlation with interpolated viral loads and the range of true correlation value at each STP and citywide.**

| STP Name | $\rho_{c\tilde{v}}$ | $\eta^{\star}$ | 95% CI for $\rho_{cv}$ |
|---|---|---|---|
| Agaram (35 MLD) | 0.91 | 3 | 0.54-1.00 |
| Bellandur (90 MLD) | 0.80 | 4 | 0.34-1.00 |
| Chikkabanavara (5 MLD) | 0.92 | 1 | 0.56-1.00 |
| Chikkabegur (5 MLD) | 0.96 | 7 | 0.66-1.00 |
| Cubbon Park (4 MLD) | 0.98 | 5 | 0.70-1.00 |
| Doddabele (60 MLD) | 0.89 | 3 | 0.50-1.00 |
| Halasuru (2 MLD) | 0.87 | 3 | 0.47-1.00 |
| Hebbal (100 MLD) | 0.82 | 5 | 0.38-1.00 |
| Hulimavu (10 MLD) | 0.92 | 3 | 0.55-1.00 |
| Jakkur (15 MLD) | 0.79 | 7 | 0.33-1.00 |
| K & C Valley (218 MLD) | 0.97 | 2 | 0.67-1.00 |
| K & C Valley (60 MLD) | 0.94 | -1 | 0.61-1.00 |
| K R Puram Old (20 MLD) | 0.97 | 6 | 0.69-1.00 |
| Kadabeesanahalli (50 MLD) | 0.94 | 3 | 0.59-1.00 |
| Kadugodi (6 MLD) | 0.93 | 6 | 0.58-1.00 |
| Kempambudhi (1 MLD) | 0.88 | 1 | 0.47-1.00 |
| Kengeri (60 MLD) | 0.99 | 12 | 0.75-0.99 |
| Lalbagh (1.5 MLD) | 0.79 | 0 | 0.32-1.00 |
| Mailasandra (75 MLD) | 0.93 | 3 | 0.58-1.00 |
| Mallathahalli (5 MLD) | 0.94 | 11 | 0.59-1.00 |
| Nagasandra (40 MLD) | 0.97 | 1 | 0.67-1.00 |
| Rajacanal (80 MLD) and Horamavu (20 MLD) | 0.82 | 5 | 0.37-1.00 |
| Sarakki (5 MLD) | 0.84 | 0 | 0.40-1.00 |
| V Valley (330 MLD) | 0.95 | 2 | 0.63-1.00 |
| Yelahanka (10 MLD) | 0.48 | 2 | 0.00-0.88 |
| Yellamallappachetty (15 MLD) | 0.88 | -4 | 0.48-1.00 |
| Bengaluru | 0.97 | 3 | 0.67-1.00 |

a $\rho_{cv}$ (resp. $\rho_{c\tilde{v}}$) is the correlation value between cases and viral load (resp. interpolated viral load).

interpolation method. The interpolation is needed as the data is available once in seven days and we use a quadratic-interpolation to follow the rising or falling trends closely without introducing a lead bias.

To detect a change-point in distribution, we have employed the CUSUM method (Fig 4). CUSUM is asymptotically optimal as long as the data is independent. We have used alternate slopes as our data sequence instead of consecutive values to ensure independence, and this modeling technique provides an advantage over other approaches, for, e.g., [12]. The change-points for the two sequences differ by at most one day. However working with differences introduces a limitation that there are fewer samples. False alarm rate and power are the parameters of interest for a fixed duration hypothesis testing problem. We have here a sequential hypothesis testing problem. False alarm rate's analogue is the average run length to false alarm (ARL). The analogue of power in the sequential setting is average detection delay (ADD). For a given ARL, the sequential CUSUM approach provides the advantage of accumulating evidence over time to have lower ADD (compared to other approaches that do not accumulate evidence). Just as the power for various false alarm rates, obtained by varying the thresholds, gives the receiver operating characteristics, Table 4 provides the relative ADD for various ARLs obtained by varying the threshold.

**Table 3. Comparison of change-points from cases and viral loads.**

| STP name | Cases CP | VCPD | VCPC | Lead-VCPD | Lead-VCPC |
|---|---|---|---|---|---|
| Jakkur (15 MLD) | 2021-12-30 | 2021-12-24 | 2021-12-24 | 6 | 6 |
| Hebbal (100 MLD) | 2021-12-28 | 2021-12-24 | 2021-12-24 | 4 | 4 |
| Rajacanal (80 MLD) & Horamavu (20 MLD) | 2021-12-26 | 2021-12-24 | 2021-12-24 | 2 | 2 |
| Yelahanka (10 MLD) | 2021-12-26 | 2021-12-24 | 2021-12-24 | 2 | 2 |
| Halasuru (2 MLD) | 2021-12-30 | 2021-12-24 | 2021-12-29 | 6 | 1 |
| Agaram (35 MLD) | 2021-12-29 | 2021-12-27 | 2021-12-29 | 2 | 0 |
| K R Puram Old (20 MLD) | 2022-01-05 | 2021-12-30 | 2022-01-05 | 6 | 0 |
| Yellamallappachetty (15 MLD) | 2021-12-28 | 2021-12-28 | 2021-12-28 | 0 | 0 |
| Cubbon Park (4 MLD) | 2021-12-28 | 2021-12-28 | 2021-12-29 | 0 | -1 |
| Lalbagh (1.5 MLD) | 2021-12-28 | 2021-12-24 | 2021-12-29 | 4 | -1 |
| Sarakki (5 MLD) | 2022-01-03 | 2021-12-24 | 2022-01-04 | 10 | -1 |
| Chikkabanavara (5 MLD) | 2021-12-24 | 2021-12-26 | 2021-12-27 | -2 | -3 |
| Chikkabegur (5 MLD) | 2022-01-03 | 2021-12-24 | 2022-01-06 | 10 | -3 |
| Mallathahalli (5 MLD) | 2022-01-03 | 2021-12-24 | 2022-01-06 | 10 | -3 |
| Nagasandra (40 MLD) | 2022-01-01 | 2021-12-26 | 2022-01-04 | 6 | -3 |
| Kadabeesanahalli (50 MLD) | 2021-12-30 | 2021-12-28 | 2022-01-03 | 2 | -4 |
| Doddabele (60 MLD) | 2022-01-01 | 2022-01-05 | 2022-01-06 | -4 | -5 |
| Kengeri (60 MLD) | 2022-01-01 | 2021-12-24 | 2022-01-06 | 8 | -5 |
| K & C Valley (218 MLD) | 2021-12-28 | 2021-12-28 | 2022-01-03 | 0 | -6 |
| Kadugodi (6 MLD) | 2021-12-30 | 2022-01-05 | 2022-01-05 | -6 | -6 |
| Bellandur (90 MLD) | 2021-12-28 | 2021-12-24 | 2022-01-04 | 4 | -7 |
| Mailasandra (75 MLD) | 2021-12-28 | 2022-01-05 | 2022-01-06 | -8 | -9 |
| V Valley (330 MLD) | 2021-12-28 | 2021-12-24 | 2022-01-06 | 4 | -9 |
| Kempambudhi (1 MLD) | 2021-12-28 | 2022-01-07 | 2022-01-07 | -10 | -10 |
| Hulimavu (10 MLD) | 2021-12-28 | 2022-01-05 | 2022-01-11 | -8 | -14 |
| K & C Valley (60 MLD) | 2021-12-24 | 2022-01-13 | 2022-01-17 | -20 | -24 |
| Bengaluru | 2021-12-29 | 2021-12-29 | NA | 0 | NA |

[a] Description of the columns. Cases CP: Change-point based on cases, VCPD: Change-point based on interpolated daily viral load, VCPC: Change-point based on sample collection date. Lead is measured in days.

[b] The alarm from cases at K & C Valley (60 MLD) is a false alarm caused by a blip in the reported cases. We notice that cases remain flat after the alarm by manual inspection.

[c] The alarms from cases for Lalbagh and Yelahanka are also false alarms. The two STPs have a negative post-change rate (see footnote of S1 Table).

[d] CUSUM is run with threshold $h = 20$.

No robust lead times were observed from viral load data over cases both at the STP level and citywide (Table 3 and Fig 5). This could be due to many reasons. The surveillance systems were already on alert for Omicron and there was good detection of cases due to the availability of RT-PCR testing. The sampling of wastewater was done once a week [12]. This infrequent sampling of wastewater from the STPs will miss viral load changes in small time windows and it is possible that early warnings are missed. Lead is also influenced by the preparedness of the public health response, by new variants, and changes in viral shedding [26,27]. Lamba et al. [12] conducted a citywide correlation analysis for a different period from January to June 2022 and found a lead of 7 days. However, this may be a due to limited testing among other factors, see [12].

**Table 4. Mean and standard deviation of lead times (in days) for VCPD and VCPC at various STPs. The distribution is over threshold _h_ ranging from 10 to 20.**

| STP Name | Lead-VCPD | Lead-VCPC |
|---|---|---|
| Yelahanka (10 MLD) | 10.55 (10.72) | 7.36 (10.01) |
| Sarakki (5 MLD) | 10.36 (2.80) | 4.36 (6.50) |
| Halasuru (2 MLD) | 6.73 (2.30) | 1.55 (3.73) |
| Hebbal (100 MLD) | 3.27 (1.54) | 1.09 (2.87) |
| Mallathahalli (5 MLD) | 6.91 (11.55) | -0.45 (13.32) |
| Jakkur (15 MLD) | -1.45 (7.09) | -2.73 (7.88) |
| Kadugodi (6 MLD) | -4.36 (5.77) | -4.36 (5.77) |
| Agaram (35 MLD) | -1.09 (4.62) | -4.55 (6.04) |
| Yellamallappachetty (15 MLD) | -3.09 (3.75) | -5.36 (4.94) |
| Rajacanal (80 MLD) and Horamavu (20 MLD) | -5.27 (5.34) | -6.00 (6.21) |
| Nagasandra (40 MLD) | 0.91 (6.29) | -6.82 (4.91) |
| Kadabeesanahalli (50 MLD) | -3.09 (6.84) | -7.40 (3.44) |
| Bellandur (90 MLD) | 2.00 (6.55) | -7.55 (9.18) |
| Cubbon Park (4 MLD) | -1.64 (2.38) | -8.09 (4.03) |
| K R Puram Old (20 MLD) | -4.55 (11.63) | -8.60 (10.36) |
| Doddabele (60 MLD) | -7.64 (4.25) | -8.64 (4.25) |
| K & C Valley (218 MLD) | -3.82 (3.46) | -9.82 (3.46) |
| Chikkabanavara (5 MLD) | -8.55 (6.10) | -10.64 (6.97) |
| Lalbagh (1.5 MLD) | -5.09 (5.42) | -10.82 (5.94) |
| Chikkabegur (5 MLD) | 1.45 (8.09) | -11.55 (8.09) |
| Kempambudhi (1 MLD) | -11.64 (5.71) | -12.09 (5.33) |
| Hulimavu (10 MLD) | -10.55 (5.12) | -15.18 (7.00) |
| Mailasandra (75 MLD) | -12.36 (5.17) | -15.18 (6.35) |
| Kengeri (60 MLD) | -5.82 (7.31) | -17.00 (7.07) |
| V Valley (330 MLD) | -4.55 (6.33) | -17.55 (6.33) |
| K & C Valley (60 MLD) | -21.56 (2.95) | -24.29 (0.70) |

During the study period, there were no mobility restrictions and other non-pharmaceutical interventions (such as wearing masks and quarantine measures) were recommended but not mandatory. We further assumed that there is a negligible floating population and sufficient homogenization. Under these assumptions an increase in viral load in an STP will correspond to an increase in cases in the corresponding catchment area. Our results are based only on this premise. A unique feature of our work is the mapping of catchment area of each STP in Bengaluru. This helped correlate daily cases at the ward level with viral loads at respective STPs. We were not able to gather such granular data for daily cases in catchment areas along with corresponding STP viral loads in other studies. For example, cases at the catchments are not reported for California's STPs. Thus, we were limited in providing a comparative analysis. Coverage of STPs in Bengaluru is extensive, capturing wastewater from a large proportion of the population. However, some portion of wastewater remains untreated and is discharged into open drains. The study by [48] examined viral loads in such open drains, providing complementary insights.

Large sewage treatment plants (STPs), like K&C Valley, cover a wide area and mix wastewater from many neighborhoods, making it harder to detect localized outbreaks. This reduces the effectiveness of wastewater surveillance for specific areas. To improve this, we suggest using sentinel manholes in key neighborhoods or high-density areas. Sampling from these manholes would allow for better detection of local viral trends.

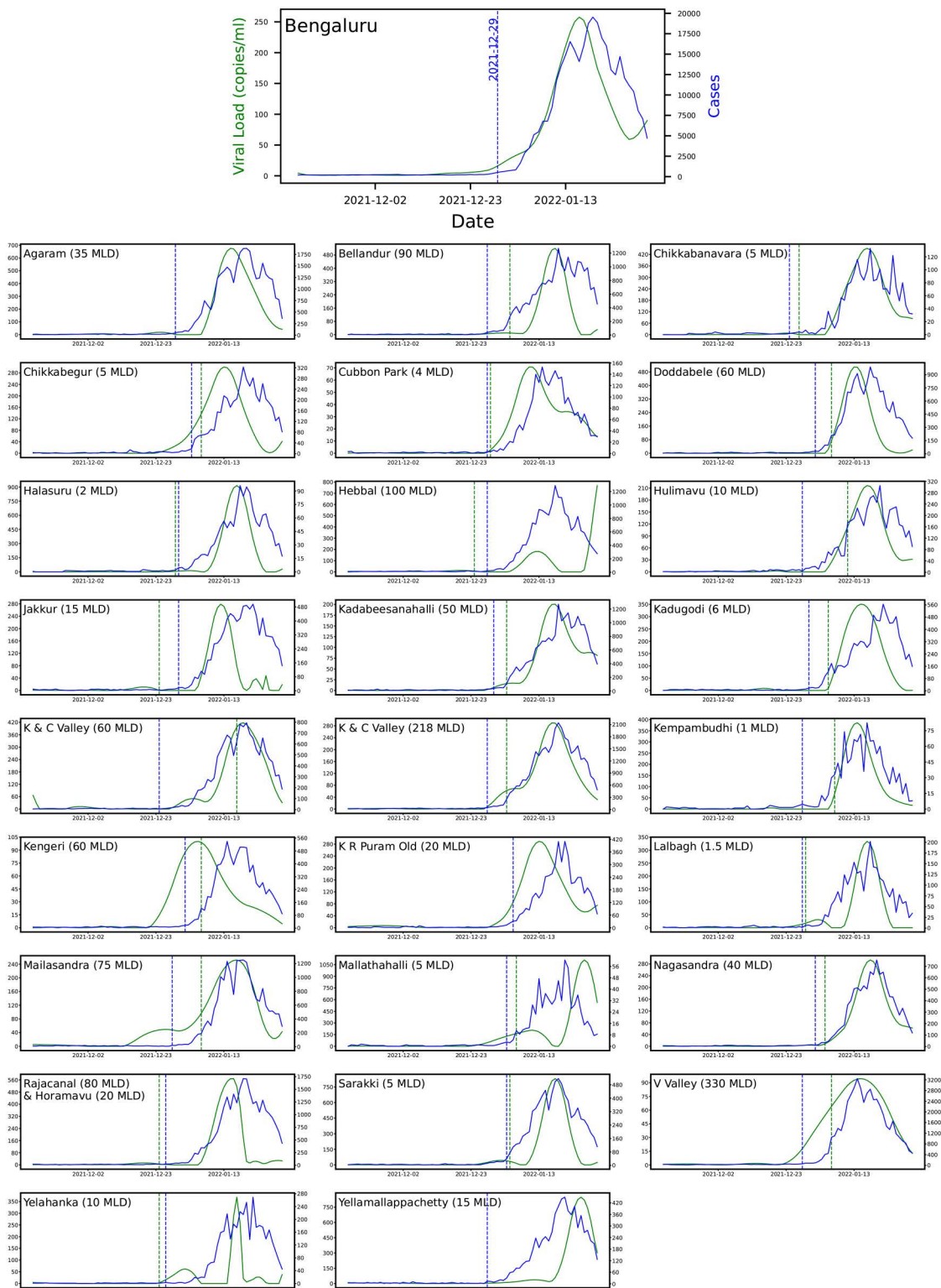

**Fig 5. A panel showing the rise of viral loads and cases citywide (Bengaluru) and at 26 STPs.** The alarm dates for each STP are indicated with dotted lines (from Table 3). The blue and the green dotted lines indicate Cases-CP and Lead-VCPC, respectively. The *x*-axis is shared across the plots.

We compute the citywide viral load by weighted average of viral loads of STPs, weighted by capacities of STPs. A more accurate method to compute the weighted viral load would be to use daily sewage flow data instead of capacities. However, reliable flow data from BWSSB is not available during the Omicron period.

Our manual mapping of wards to STPs, based on inspection of the sewerage network and expert consultation, does not easily scale to other cities. It may be possible to automatically map regions to STPs using the sewerage network and the city's topography. Given a detailed network file, such as a GIS file, and elevation data, flow directions can be traced by following pipe connections and slopes, since wastewater flows downhill. This would allow automated assignment of areas, such as wards, to their respective STPs. While practical challenges like missing data, pumping stations, or undocumented connections may affect accuracy, in principle, accurate and updated network and topographic data would enable systematic and reliable mapping. This approach can be explored as future work to improve the precision and scalability of wastewater surveillance efforts.

In summary, as the viral data was limited to 8–9 samples per STP, we have used interpolation to get the daily viral load estimates. We have quantified the consequent uncertainty in correlation with daily cases and provided error bounds. Even with these limitations our analysis during the Omicron wave in Bengaluru suggests that an increase in wastewater viral loads correlates with increase in COVID-19 cases. We also observed this correlation at individual STP levels. Consequently, we conclude that in the absence of testing, continuous wastewater surveillance can help in the monitoring of SARS-CoV2 circulation. However, wastewater surveillance did not give a reliable early warning during the Omicron wave. The frequency of sampling may be increased to daily sampling during periods of infection to enable timely interventions. A more detailed mapping of the sewersheds, e.g., via satellite imaging, can enhance the surveillance area.

## Supporting information

**S1 Fig. Drainage pipes from wards leading to the Hebbal STP.**
(PDF)

**S2 Fig. BWSSB sewer lines coverage.**
(PDF)

**S3 Fig. A scatter plot of viral loads and cases from four STPs.**
(PDF)

**S4 Fig. Error bars for viral load estimates at STPs in California.**
(PDF)

**S5 Fig. Error bars for cases estimates at STP catchments in Bengaluru.**
(PDF)

**S6 Fig. Distribution of correlation values between a vector and its noisy version.**
(PDF)

**S1 Table. Parameter values of the pre-and-post distributions of alternate slopes.**
(PDF)

**S2 Table. Correlation between daily viral loads and values interpolated from weekly samples in California.**
(PDF)

**S3 Table. Correlation between daily cases and interpolated cases from weekly samples in Bengaluru.**
(PDF)

**S4 Table. List of files related to datasets, code, and results.**
(PDF)

**S1 Text. An example of computing weighted viral load.**
(PDF)

**S2 Text. A summary of our work in Kannada.**
(PDF)

## Acknowledgments

We gratefully acknowledge the support received from BWSSB. We thank Dr. Bhaskar Rajakumar for helping us coordinate with BBMP. We gratefully acknowledge the contribution of Jagadish Midthala towards the conceptualization, data curation, investigation, methodology, software, and in the preparation of the manuscript.

## Author contributions

**Conceptualization:** Rajesh Sundaresan.

**Data curation:** Farah Ishtiaq, Tarun Khandelwal.

**Investigation:** Siva Athreya, Tarun Khandelwal, Rajesh Sundaresan.

**Methodology:** Siva Athreya, Farah Ishtiaq, Tarun Khandelwal, Chitra Pattabiraman, Lakshminarayana Rao, Rajesh Sundaresan, Reshma Mohan Thattaramppilly.

**Project administration:** Rajesh Sundaresan.

**Resources:** Chitra Pattabiraman, Lakshminarayana Rao.

**Software:** Tarun Khandelwal.

**Supervision:** Siva Athreya, Chitra Pattabiraman, Rajesh Sundaresan.

**Visualization:** Siva Athreya, Tarun Khandelwal, Chitra Pattabiraman.

**Writing – original draft:** Chitra Pattabiraman.

**Writing – review & editing:** Siva Athreya, Farah Ishtiaq, Tarun Khandelwal, Chitra Pattabiraman, Lakshminarayana Rao, Rajesh Sundaresan, Reshma Mohan Thattaramppilly.

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
