## [Decision Letter · Decision Letter 0]

10 Oct 2024

PGPH-D-24-02064

Localized wastewater surveillance showed correlation but no early warning during Bengaluru's Omicron wave

Dear Dr. Midthala,

Thank you for submitting your manuscript to PLOS Global Public Health. After careful consideration, we feel that it has merit but does not fully meet PLOS Global Public Health’s publication criteria as it currently stands. Therefore, we invite you to submit a revised version of the manuscript that addresses the points raised during the review process.

The reviewers have raised several important questions related to the limited sample number and the generalization of the results. Please address the points raised by the reviewers carefully and resubmit the modified manuscript.

We look forward to receiving your revised manuscript.

Kind regards,

Mohan Amarasiri

Academic Editor

Journal Requirements:

1. We ask that a manuscript source file is provided at Revision. Please upload your manuscript file as a .doc, .docx, .rtf or .tex.

2. Figure 1, 2, 3 and 4: please (a) provide a direct link to the base layer of the map (i.e., the country or region border shape) and ensure this is also included in the figure legend; and (b) provide a link to the terms of use / license information for the base layer image or shapefile. We cannot publish proprietary or copyrighted maps (e.g. Google Maps, Mapquest) and the terms of use for your map base layer must be compatible with our CC-BY 4.0 license.

Additional Editor Comments (if provided):

Reviewers' comments:

Reviewer's Responses to Questions

**Comments to the Author**

1. Does this manuscript meet PLOS Global Public Health’s publication criteria? Is the manuscript technically sound, and do the data support the conclusions? The manuscript must describe methodologically and ethically rigorous research with conclusions that are appropriately drawn based on the data presented.? Is the manuscript technically sound, and do the data support the conclusions? The manuscript must describe methodologically and ethically rigorous research with conclusions that are appropriately drawn based on the data presented.

Reviewer #1: Partly

Reviewer #2: Partly

2. Has the statistical analysis been performed appropriately and rigorously?

Reviewer #1: No

Reviewer #2: Yes

3. Have the authors made all data underlying the findings in their manuscript fully available (please refer to the Data Availability Statement at the start of the manuscript PDF file)?

The PLOS Data policy requires authors to make all data underlying the findings described in their manuscript fully available without restriction, with rare exception. The data should be provided as part of the manuscript or its supporting information, or deposited to a public repository. For example, in addition to summary statistics, the data points behind means, medians and variance measures should be available. If there are restrictions on publicly sharing data—e.g. participant privacy or use of data from a third party—those must be specified.requires authors to make all data underlying the findings described in their manuscript fully available without restriction, with rare exception. The data should be provided as part of the manuscript or its supporting information, or deposited to a public repository. For example, in addition to summary statistics, the data points behind means, medians and variance measures should be available. If there are restrictions on publicly sharing data—e.g. participant privacy or use of data from a third party—those must be specified.

Reviewer #1: Yes

Reviewer #2: Yes

4. Is the manuscript presented in an intelligible fashion and written in standard English?

Reviewer #1: Yes

Reviewer #2: Yes

Reviewer #1: This manuscript presents a study analyzing correlations between SARS-CoV-2 viral loads in wastewater and COVID-19 case numbers in Bengaluru, India, during the Omicron wave. The authors examine data at both citywide and localized levels across 26 sewage treatment plants (STPs) from December 15, 2021, to January 15, 2022. The study addresses an important topic and offers some interesting insights, but there are several limitations that need to be addressed to strengthen the manuscript.

One major concern is the limited number of viral load data points available. With weekly sampling over a one-month period, there are only 4-5 data points per STP. This small sample size may impact the reliability of the correlation analyses that are central to the study. The authors should consider providing a power analysis to assess if this sample size is sufficient for drawing robust conclusions. If not, the findings and claims may need to be qualified accordingly. The authors should discuss the implications of having few data points, including the potential impact on the uncertainty of correlation estimates, sensitivity to outliers, and statistical power. Limitations related to sample size should be addressed in the discussion.

Another issue is the use of quadratic interpolation to estimate daily viral loads from the weekly samples. The reliability of this interpolation approach should be further validated, and uncertainties related to the interpolated values could be quantified. The authors may consider a sensitivity analysis using the raw weekly data to evaluate the impact of interpolation on the results.

Given the limited data, some conclusions drawn from the correlation analyses may need to be tempered. The authors should consider focusing conclusions on this specific case study and avoid broad generalizations. Claims could be qualified by emphasizing the uncertainties arising from data limitations.

The one-month study period may also constrain the generalizability of the findings. The authors could contextualize how focusing on this specific window within the Omicron wave may impact results compared to examining viral load and case correlations over longer timeframes. If feasible, expanding the analysis to cover multiple COVID waves could strengthen the study.

Additionally, the manuscript could benefit from a discussion of potential confounding factors that may influence viral load and case correlations, such as changes in control measures and human mobility patterns over time. Comparing findings to studies of SARS-CoV-2 wastewater surveillance from other regions could provide useful context, even if a full comparative analysis is beyond the current scope.

In summary, this study contributes to the growing body of research on wastewater surveillance for COVID-19. However, the conclusions that can be drawn are limited by the small number of data points. Revisions addressing and discussing the implications of these limitations could strengthen the robustness of the findings. I recommend that the authors consider the suggestions above to improve the manuscript. With appropriate revisions, this study could provide a valuable insight into the relationship between wastewater viral loads and COVID-19 cases during the Omicron wave in Bengaluru, while acknowledging the constraints posed by the available data.

Reviewer #2: The authors wrote about the localized wastewater surveillance and their application in early warning for SARS-CoV-2. The manuscript needs to have more discussion but overall, the model is interesting.

Abstract

Please be aware that SARS-CoV-2 is the name of the virus. Covid 19 is the name of the disease. Check the whole document and fix if there are mistakes.

Introduction

Line 3: poliovirus, not Poliovirus.

Line 4: antimicrobial resistance gene? Bacteria?

Line 5: Not all virus is an RNA virus.

Line 21: … source tracking. What source tracking?

Line 31-32: Lead time of 2 days and as long as 1-2 weeks. Can we use this as a guidance? Check the result and see if the model showed at least 2 days lead with a good correlation with clinical cases.

Line 35: Please specify what virus. SARS-CoV-2?

Method

Line 65-66: Please specify why grab method was chosen, what are their limitation, around what time those samples were collected?

How did the author measure the RNA copies of SARS-CoV-2? qPCR? What are the primers and probes, or kit used? What PCR machine was used? What is the limit of detection? What is the positive control used?

Line 105-121: Did the author try to normalize the data by using flow and population? Please try and see if it works better with the model.

What is quadratic interpolation and how did the author did it? What did author mean by clip any negative result? Cutting the data or use 0? Why don’t use a limit of detection number?

Line 117-121: How did the author combine 2 STP and replace their daily viral loads with weighted average?? Please describe 1 calculation as example.

For clinical cases. Was it the data based on hospital report? Or based on patient residency?

Line 148-159: CUSUM. Please mention the full words before abbreviating it.

How to measure the average run length? Expected time until a change was detected? Is it the same as what mentioned at line 157?

So what was the h-value that applied here?

Results

Table 1 is too long. Maybe can be mentioned as supplementary information.

Please make a summary of table 1. Like how many wards and their range of MLD.

Line 182: Correlation analysis was not mentioned in the methodology section.

Why was Pearson analysis considered?

Line 184: This should be in method section

Table 2 and 3: How the author concluded that there is no lead? there are lead number with good correlation there.

Figure 5: How was the epi curve data obtained. It should be mentioned at the method section.

How to determine the case shift?

Figure 6: This should be discussed in the discussion section. How did the data compared based on windows of size?

Line 198-219: Need to be in method section. Why did the author explain analysis method at the result section?

Just curious, if the viral loads were measured weekly, is it possible to use the weekly average clinical cases for CUSUM analysis? If yes, why it was not conducted? Is it because the window is too small and it will be hard to get correlation?

Based on Figure 9, I can see there was early warning where the peak was detected after the VCPD.

Discussion.

In my opinion, the discussion part is still needing more works.

Line 228-231: Just repetition from method.

Line 234-243: Early warning is not well defined. If the peak from wastewater followed by peak from clinical cases after several days, does not it count as early warning? Based on Fig 9 I can see that. What is the threshold for early warning? Figure 8 says 4 days. Please discuss from figure 9, what was the range for VDPC for all the STPs, did it consider as early warning. If not, why?

Author only mentioned there is a good correlation between the wave of wastewater and clinical but no further discussion. Why are there good correlation? Is it the model? If yes how about comparing the model used here with another model? Please find more literature to discuss. For example, please compare to ARIMA model.

An increase in wastewater viral load does not necessarily an outbreak indicator.

**Do you want your identity to be public for this peer review?** For information about this choice, including consent withdrawal, please see our Privacy Policy..

Reviewer #1: No

Reviewer #2: No

---

## [Decision Letter · Decision Letter 1]

8 Apr 2025

PGPH-D-24-02064R1

Localized wastewater surveillance showed correlation but no early warning during Bengaluru's Omicron wave

Dear Dr. Midthala,

Thank you for submitting your manuscript to PLOS Global Public Health. After careful consideration, we feel that it has merit but does not fully meet PLOS Global Public Health’s publication criteria as it currently stands. Therefore, we invite you to submit a revised version of the manuscript that addresses the points raised during the review process.

One reviewer has highlighted the necessity to improve the methodology section.

We look forward to receiving your revised manuscript.

Kind regards,

Mohan Amarasiri

Academic Editor

Journal Requirements:

Additional Editor Comments (if provided):

Reviewers' comments:

Reviewer's Responses to Questions

**Comments to the Author**

Reviewer #1: All comments have been addressed

Reviewer #2: All comments have been addressed

publication criteria? Is the manuscript technically sound, and do the data support the conclusions? The manuscript must describe methodologically and ethically rigorous research with conclusions that are appropriately drawn based on the data presented.? Is the manuscript technically sound, and do the data support the conclusions? The manuscript must describe methodologically and ethically rigorous research with conclusions that are appropriately drawn based on the data presented.

Reviewer #1: Yes

Reviewer #2: Partly

3. Has the statistical analysis been performed appropriately and rigorously?

Reviewer #1: Yes

Reviewer #2: Yes

4. Have the authors made all data underlying the findings in their manuscript fully available (please refer to the Data Availability Statement at the start of the manuscript PDF file)?

The PLOS Data policy requires authors to make all data underlying the findings described in their manuscript fully available without restriction, with rare exception. The data should be provided as part of the manuscript or its supporting information, or deposited to a public repository. For example, in addition to summary statistics, the data points behind means, medians and variance measures should be available. If there are restrictions on publicly sharing data—e.g. participant privacy or use of data from a third party—those must be specified.requires authors to make all data underlying the findings described in their manuscript fully available without restriction, with rare exception. The data should be provided as part of the manuscript or its supporting information, or deposited to a public repository. For example, in addition to summary statistics, the data points behind means, medians and variance measures should be available. If there are restrictions on publicly sharing data—e.g. participant privacy or use of data from a third party—those must be specified.

Reviewer #1: Yes

Reviewer #2: Yes

5. Is the manuscript presented in an intelligible fashion and written in standard English?

Reviewer #1: Yes

Reviewer #2: Yes

Reviewer #1: I recommend this manuscript for publication in its current form. The authors have thoroughly addressed all major methodological concerns through expanded analysis, validation, and appropriate uncertainty quantification. The conclusions are well-supported and properly qualified, making a valuable contribution to understanding localized wastewater surveillance during the Omicron wave.

Reviewer #2: Reviewer Comments on the Rebuttal Manuscript

The author has submitted a revised version of the manuscript in response to previous feedback. The revised manuscript is better written and demonstrates improved clarity. I have some comments remaining for further revision:

Abstract

Please include the correlation value(s) referenced.

Kindly remove the statement mentioning who conducted the surveillance in 2021, as it is not relevant here.

Please clarify what is meant by “correspond” in the context used.

Author Summary

The first bullet point is not a result of this study and should therefore be removed.

Introduction

The explanation of viral load currently in line 46 should be moved earlier, to line 41, for better context.

Please clearly state the aims of the study in the final paragraph of the introduction. Refer to line 63 of the methodology for guidance.

Methodology

As noted for the abstract, please remove the sentence on line 69 referencing who conducted the surveillance.

Line 69: What was the normalization method used? Please specify.

Line 73: The researcher’s name mentioned here should instead be included in the Acknowledgments section.

Figures 3 and 4 could be moved to the supplementary material. Figures 1 and 2 are sufficient for illustrating the map.

Lines 128–132: Please include only the formula; the example calculation can be moved to the supplementary section.

When describing the correlation analysis, please explain the assumption of linearity between viral load and COVID-19 case numbers. Why is a linear relationship appropriate here?

Figure 5 should be moved to the supplementary material. Results should not be presented in the Methodology section.

The explanation of the interpolation method should be made more concise. The associated calculation results (Figures 6 and 7, Tables 1 and 2) should be moved to the supplementary section, as these are results.

The same applies to the CUSUM analysis: Figures 8 and Table 3 should be relocated to the supplementary material.

For Table 5, please define what "PCv" refers to.

Discussion

The interpolation method used for a small dataset is interesting. Please discuss the interpretation of the results presented in Figures 6 and 7. What does the error factor represent? What values are considered good or poor in this context? How small can the dataset be for this interpolation method to still be effective?

Please include a discussion of the study’s limitations and suggestions for improvement in future work.

Line 393: Is the observation discussed here possibly due to limitations in the sampling strategy or data availability? Would more complete data potentially lead to different outcomes?

**Do you want your identity to be public for this peer review?** For information about this choice, including consent withdrawal, please see our Privacy Policy..

Reviewer #1: **Yes:** Charin ModchangCharin ModchangCharin ModchangCharin Modchang

Reviewer #2: No

---

## [Editor Report · Decision Letter 2]

15 Aug 2025

PGPH-D-24-02064R2

Localized wastewater surveillance showed correlation but no early warning during Bengaluru's Omicron wave

Dear Dr. Midthala,

Thank you for submitting your manuscript to PLOS Global Public Health.

In accordance with your request, we are returning the manuscript to your account so that you may revise it in light of the disparity in viral load values reported in your submission and in doi: 10.1016/j.lansea.2025.100619.

A rebuttal letter explaining the changes made to the manuscript.A marked-up copy of your manuscript that highlights changes made to the original version. You should upload this as a separate file labeled 'Revised Manuscript with Track Changes'.An unmarked version of your revised paper without tracked changes. You should upload this as a separate file labeled 'Manuscript'.

We look forward to receiving your revised manuscript.

Kind regards,

Steve Zimmerman, PhD

PLOS Staff Editor
---

## [Decision Letter · Decision Letter 3]

21 Nov 2025

PGPH-D-24-02064R3

Localized wastewater surveillance showed correlation but no early warning during Bengaluru's Omicron wave

Dear Dr. Rajesh,

Thank you for submitting your manuscript to PLOS Global Public Health. After careful consideration, we feel that it has merit but does not fully meet PLOS Global Public Health’s publication criteria as it currently stands. Therefore, we invite you to submit a revised version of the manuscript that addresses the points raised during the review process.

We look forward to receiving your revised manuscript.

Kind regards,

Mohan Amarasiri

Academic Editor

Journal Requirements:

Additional Editor Comments (if provided):

Reviewers' comments:

Reviewer's Responses to Questions

**Comments to the Author**

Reviewer #1: All comments have been addressed

Reviewer #2: All comments have been addressed

publication criteria? Is the manuscript technically sound, and do the data support the conclusions? The manuscript must describe methodologically and ethically rigorous research with conclusions that are appropriately drawn based on the data presented.? Is the manuscript technically sound, and do the data support the conclusions? The manuscript must describe methodologically and ethically rigorous research with conclusions that are appropriately drawn based on the data presented.

Reviewer #1: Yes

Reviewer #2: Yes

3. Has the statistical analysis been performed appropriately and rigorously?

Reviewer #1: Yes

Reviewer #2: Yes

4. Have the authors made all data underlying the findings in their manuscript fully available (please refer to the Data Availability Statement at the start of the manuscript PDF file)?

The PLOS Data policy requires authors to make all data underlying the findings described in their manuscript fully available without restriction, with rare exception. The data should be provided as part of the manuscript or its supporting information, or deposited to a public repository. For example, in addition to summary statistics, the data points behind means, medians and variance measures should be available. If there are restrictions on publicly sharing data—e.g. participant privacy or use of data from a third party—those must be specified.requires authors to make all data underlying the findings described in their manuscript fully available without restriction, with rare exception. The data should be provided as part of the manuscript or its supporting information, or deposited to a public repository. For example, in addition to summary statistics, the data points behind means, medians and variance measures should be available. If there are restrictions on publicly sharing data—e.g. participant privacy or use of data from a third party—those must be specified.

Reviewer #1: Yes

Reviewer #2: Yes

5. Is the manuscript presented in an intelligible fashion and written in standard English?

Reviewer #1: Yes

Reviewer #2: Yes

Reviewer #1: The authors have fully and transparently resolved the data discrepancy and other issues raised, without affecting the study’s conclusions. I recommend acceptance of the manuscript.

Reviewer #2: This manuscript addresses a timely and important topic in covid19 surveillance by examining the relationship between wastewater viral loads and clinical case trends across multiple sites. As wastewater-based epidemiology continues to evolve as a complementary tool for infectious disease monitoring, especially for enteric and respiratory viruses, studies that evaluate its accuracy, temporal dynamics, and analytic approaches are essential. The authors present relevant data with clear public health implications; however, several methodological clarifications, expanded discussion points, and additional statistical context are required to enhance the rigor, interpretability, and impact of the findings. The comments below are intended to strengthen the manuscript and support its suitability for publication in PLOS Pathogens.

Abstract

Please include the confidence-interval values for city-level clinical data versus aggregated viral load so that the similarity with localized viral load trends can be justified.

Introduction

Line 4: Please revise this sentence. Antimicrobial resistance (AMR) and antimicrobial resistance genes (ARG) surveillance represents a recent application of wastewater surveillance. It should not be presented as a previous or early use of environmental surveillance. The cited reference is correct; however, the interpretation requires adjustment.

Line 29: Please define the criteria used to classify a “strong correlation.”

Line 33: Please elaborate on the reasons why increased sampling frequency strengthens correlation analysis.

Line 34: Please provide a clear definition of “lead times.”

Methods

Lines 118–119: Please clarify the Limit of Detection (LoD) and Limit of Quantification (LoQ). If the LoD values are available, please explain why they were not used in place of zero. In addition, please clarify whether applying the LoD threshold would alter the results of the correlation analysis.

Lines 122–124: Please explain why the sewage treatment plant (STP) capacity was used rather than daily or weekly flow-rate averages.

Figure 4: Please include the units for viral load. In addition, the final portion of the figure appears to show decreasing clinical cases while viral loads increase. This pattern appears inconsistent with other STPs shown in Figure 5. Please discuss any potential explanations.

Discussion

Lines 343–349: I recommend expanding this section by introducing standard correlation strength categories (for example, low, moderate, and high) and discussing the rationale behind these classifications. Please also describe the statistical methods used, why they are appropriate, and whether they have been commonly applied in similar studies. Highlight which STP demonstrated the highest and lowest correlations and consider providing potential explanations, such as geographic context or differences in data availability.

Lines 355–357: Please clarify the reason for the low viral loads observed during this period. Review lines 420–422 and assess whether the reported error bounds are sufficiently narrow to support confidence in the findings.

Lines 360–362: Please justify the interpolation method used. A brief explanation regarding why interpolation is appropriate for this dataset would strengthen this section.

Lines 363–372: This portion reads more like a methods description rather than a discussion. Please revise to address the advantages of the modelling approach, cite comparable studies that have applied similar models, and discuss alternative models that could have been used, including relative strengths and limitations.

**Do you want your identity to be public for this peer review?** For information about this choice, including consent withdrawal, please see our Privacy Policy..

Reviewer #1: No

Reviewer #2: No

---

## [Editor Report · Decision Letter 4]

29 Dec 2025

Localized wastewater surveillance showed correlation but no early warning during Bengaluru's Omicron wave

PGPH-D-24-02064R4

Dear Prof Rajesh,

We are pleased to inform you that your manuscript 'Localized wastewater surveillance showed correlation but no early warning during Bengaluru's Omicron wave' has been provisionally accepted for publication in PLOS Global Public Health.

Best regards,

Mohan Amarasiri

Academic Editor